# An Improved Genetic Algorithm for Path-Planning of Unmanned Surface Vehicle

**DOI:** 10.3390/s19112640

**Published:** 2019-06-11

**Authors:** Junfeng Xin, Jiabao Zhong, Fengru Yang, Ying Cui, Jinlu Sheng

**Affiliations:** 1College of Electromechanical Engineering, Qingdao University of Science and Technology, Qingdao 266061, China; jf.xin@163.com (J.X.); zhongjbchn@163.com (J.Z.); seawolf_yfr@163.com (F.Y.); cuiying@qust.edu.cn (Y.C.); 2Transport College, Chongqing Jiaotong University, Chongqing 400074, China

**Keywords:** genetic algorithm, unmanned surface vehicle, path planning, multi-domain inversion, Monte-Carlo simulation

## Abstract

The genetic algorithm (GA) is an effective method to solve the path-planning problem and help realize the autonomous navigation for and control of unmanned surface vehicles. In order to overcome the inherent shortcomings of conventional GA such as population premature and slow convergence speed, this paper proposes the strategy of increasing the number of offsprings by using the multi-domain inversion. Meanwhile, a second fitness evaluation was conducted to eliminate undesirable offsprings and reserve the most advantageous individuals. The improvement could help enhance the capability of local search effectively and increase the probability of generating excellent individuals. Monte-Carlo simulations for five examples from the library for the travelling salesman problem were first conducted to assess the effectiveness of algorithms. Furthermore, the improved algorithms were applied to the navigation, guidance, and control system of an unmanned surface vehicle in a real maritime environment. Comparative study reveals that the algorithm with multi-domain inversion is superior with a desirable balance between the path length and time-cost, and has a shorter optimal path, a faster convergence speed, and better robustness than the others.

## 1. Introduction

The traveling salesman problem (TSP) is a typical non-deterministic polynomial (NP)hard problem with the goal of designing the shortest route for a traveler to visit each city without repetition, followed by returning to the starting city. In production and life, TSP has been widely used as a model in many fields, such as vehicle path planning [1,2,3], machine learning [4], temporal graphs [5], word sense disambiguation [6], green logistics [7], fuel efficiency management [8], wireless charging [9], and so on. Hence, solving the TSP is of great significance for household, civil, and military applications.

Recent studies tend to utilize approximate algorithms for the TSP, such as genetic algorithm (GA), stimulated annealing algorithm, ant colony algorithm, and neural network algorithm [10,11]. By contrast, GA has advantages of higher robustness and stronger global search capability, and hence has been applied to the trajectory planning problem for various kinds of autonomous equipment such as robots, unmanned aerial vehicles (UAVs), and unmanned surface vehicles (USVs). To solve the collision-free shortest path planning problem of mobile agents, Lee et al. (2018) applied the obstacle-based GA to shrink the search areas and obtain a considerable path with shorter length and time-cost [12]. Elhoseny et al. (2018) modified the conventional genetic algorithm (CGA) for a mobile robot to search for the control points of a Bezier curve and designed the shortest route in a dynamic working field [13]. In addition, Sahingoz (2013) applied the parallel GA to the multi-UAVs system in a multi-core environment. The preliminary planned path was further smoothed by a Bezier curve to generate the final flyable trajectory [14]. Ergezer et al. (2013) improved the GA with novel evolutionary operators for multi-UAVs. The favorable route was obtained with consideration of 3D environmental constraints and the maximization of the collected information from desired regions [15]. In terms of USVs, Kim et al. (2017) combined three objective functions of avoiding obstacles, reaching the target, and minimizing travel time to evaluate the fitness of the path under ocean environmental loads [16]. 

Moreover, in order to overcome the inherent issues of CGA such as slow convergence speed, poor capability of local search, and easy occurrence of premature convergence, combinations of two or more optimization algorithms based on biological evolutionary and mathematical ecological theory have been employed to improve the algorithm performance. Wang et al. (2016) improved the crossover operator to produce more offsprings, thus enriching the population diversity. With tests of several TSP examples, the multi-offspring method (MO-GA) was proven to converge faster, reaching a deeper minimum of cost function than the CGA [17]. With the multiple objectives of the minimization of total travel fuzzy cost and fuzzy time, Khanra et al. (2019) combined the ant colony optimization and GA to solve the four-dimensional imprecise TSP, which included source, destination, conveyances, and routes [18]. In the centralized UAV placement strategy proposed by Sabino et al. (2018), the elitist non-dominated sorting genetic algorithm was employed to design the optimal positions of UAVs with the consideration of ground nodes’ positions [19]. Silva et al. (2018) applied the dynamic planning navigation algorithm, which was based on GA and had better robustness and effectiveness, to the autonomous navigation of mobile terrestrial robots under unknown dynamic environments [20]. In order to solve the group trading strategy portfolio in stock markets, Chen et al. (2019) employed the grouping genetic algorithm (GGA) with a fitness function that was calculated by group balance, weight balance, portfolio return, and risk [21].

In this paper, the CGA is improved by optimizing the chromosome inversion operation. First, the double-domain inversion-based algorithm (DDIGA) is proposed with two inversion operations between four randomly sorted inversion points. Furthermore, the number of inversion domains is increased through permutation and combination of the sequence of the four inversion points. The multi-domain inversion-based algorithm (MDIGA) is supposed to further enhance the local search capability since the offsprings are significantly increased, and only the inversed chromosome with the best fitness survives and is transferred to the new generation.

The contributions of this work consist of three aspects: (1) The optimization strategy of multi-domain inversion is added after the crossover and mutation operations, with more offsprings to be selected in order to increase the probability of generating excellent individuals and avoid the population premature; (2) the MDIGA performs better in both reducing the optimal path length and enhancing the robustness than the CGA; and (3) path-planning for a USV is conducted using the MDIGA, which generates feasible routes with satisfactory length.

This paper is organized as follows. The details of CGA, DDIGA, and MDIGA are introduced in Section 2. The feasibility of MDIGA is then analyzed by comparison with other state-of-the-art approaches in Section 3. Monte-Carlo simulations for five instances from the library for TSP (TSPLIB) and the application tests to a USV are conducted to evaluate the effectiveness of every algorithm, which are described and analyzed in Section 4 and Section 5. Conclusions and future research interests are drawn in Section 6.

## 2. Proposed Algorithms

### 2.1. Conventional Genetic Algorithm

Figure 1 illustrates the computing steps of the CGA [22,23]. The symbolic coding is chosen as the encoding method, which uses the string with the sequence numbers of visiting cities to represent each chromosome. Genetic parameters, such as population size, crossover, and mutation probabilities, are normally defined by human experience. After the optimization problem is confirmed, the initial population of candidate solutions with a certain size is generated randomly. The fitness function, defined by 1/*len* (*len* stands for the relative route length of each chromosome), is used to evaluate the fitness of each individual. The fitter ones will survive for the reproduction. Then, the algorithm proceeds to improve the population through repetitive operations of crossover, mutation, and selection. The evolutionary process will be terminated if certain criteria are satisfied or if the maximum number of iterations is reached.

In CGAs, the crossover is performed to concatenate parts of two parent chromosomes, which are separated by determined break points, and generate two offsprings with a certain crossover probability (*P_C_*). Meanwhile, the mutation interchanges the positions of genes at two randomly chosen mutation points in a single chromosome. Accordingly, the mutation occurs with a certain mutation probability (*P_M_*). It should be noted that the crossover helps the population to converge by making the chromosomes alike, whereas the mutation brings the genetic diversity back into the population in case of the local optimum. In this paper, the single-domain inversion-based algorithm (SDIGA) is presented with a further inversion operation added after the mutation in the CGA. Two different genes in a single chromosome are defined as inversion points, between which the fragment is named an inversion domain. The fragment is then turned through 180° (inversed) and inserted back into the original position of the chromosome. The schematic layouts for crossover, mutation, and single-domain inversion are illustrated in Figure 2, Figure 3 and Figure 4, respectively.

### 2.2. Double-Domain Inversion-Based Genetic Algorithm

In CGA, the symbolic coding is normally used as the chromosome encoding with the crossover operator of a partially mapped crossover (PMC) to solve the TSP [24,25]. However, this crossover operator causes a terrible destruction to the parent chromosomes. Only a fraction of parent genes could survive, and most genes of offspring chromosomes are generated during the evolutionary process; this is not conducive to the inheritance of advantage genes from parent chromosomes. In addition, the mutation or the single-domain inversion has evident deficiencies of local search capacities due to their limited transformation of genes. Hence, the strategy named the double-domain inversion-based genetic algorithm (DDIGA) is introduced, as shown in Figure 1.

The positions of four different genes are randomly defined as the inversion points from the encoding string of a chromosome. Two domains are generated between the first two points and the latter two points, respectively. The fragments in both domains are inversed simultaneously to reproduce an offspring. The fitness of both child and parent chromosomes will be compared to determine the fitter one for the next generation. The double-domain inversion is illustrated in Figure 5, in which I stands for a parent chromosome, and I’ is the child chromosome after inversion.

It is supposed that introducing the double-domain inversion could help retain more advantage genes from parent chromosomes and generate more adaptive encoded strings for child chromosomes. Meanwhile, the capacity of local search may be improved since the compassion of fitness could guarantee the evolution towards a higher fitness level.

### 2.3. Multi-Domain Inversion-Based Genetic Algorithm

It is known that in the CGA, the number of generated offsprings is normally the same as the number of parent chromosomes. From the point of view of the biological theory foundation, the number of offsprings needs to be larger than the number of parents so as to prevent species extinction and maintain species diversity in the process of biological evolution [17].

As mentioned in Section 2.2, four randomly sorted points create two domains for the DDIGA, and only one child chromosome is generated after two inversions. In fact, however, every two of four inversion points could define an inversion domain. According to permutation and combination theory, six domains for a single inversion could be found in total. Hence, six extra child chromosomes would be reproduced through a single inversion of each domain in the parent chromosome; this would increase the probability of finding a fitter offspring for every generation to some degree.

Inspired by the above discussion, the MDIGA is proposed based on multi-domain inversion to increase the number of inversion domains and child chromosomes. As shown in Figure 6, four inversion points, named a–d, are randomly defined in the encoded string. Six child chromosomes I’_1_- I’_6_ are generated by a single inversion within domains a-b, a-c, a-d, b-c, b-d, and c-d, respectively. Similarly with the DDIGA, I’_7_ is generated by double inversions within domains a-b and c-d. The parent and seven child chromosomes are then sorted according to the fitness function 1/*D* (*D* is the path length of each chromosome). Only the most advantaged chromosome I’ (I’_5_ in this case) is reserved for the next generation, while the others would be eliminated completely.

Theoretically, MDIGA would accelerate the speed of evolution towards a higher fitness for the population and enhance convergence precision and robustness of algorithm.

## 3. Feasibility Analysis

This section analyzes the feasibility of improved algorithms. First, it should be noted that the idea of the MDIGA and the MO-GA in [17] are both based on the biological theory foundation of multiple offsprings and the mechanism of population competition. However, the two algorithms differ in the optimized genetic operator and encoding method. The MO-GA is based on the crossover operation and generates 2α (α ∈ {2, 3, 4, …}) offsprings for a pair of parent chromosomes, while MDIGA improves the inversion operation and generates seven offsprings for one parent chromosome. In addition, MO-GA employs binary code and MDIGA uses real-number code. Table 1 compares the planned path length of the MO-GA and the MDIGA for three TSPLIB instances: burma14, eil51, and kroB100. For the case of burma14, each algorithm can obtain a desirable solution that is exactly the same as the known optimal solution in the TSPLIB. As for eil51, the MDIGA is better than the MO-GA, with a length of 434.08 m, although the value is 1.9% larger than the known optimal value. For kroB100, the planned path of the MDIGA is 3.5% longer than those of the MO-GA and the known optimal value of the TSPLIB.

As mentioned in Section 2.1, the values of crossover probability (*P_C_*) and mutation probability (*P_M_*) are normally determined by experience. According to the suggestions by Elhoseny et al. [13], the value range of *P_C_* is suggested to be from 0.7 to 1. A lower value than this range will reduce the crossover operation, which is not efficient for evolution. Meanwhile, the value range of *P_M_* is suggested to be 0.001 to 0.05. A larger value than this range will increase the mutation operation, making the algorithm jump out of the best solution and deteriorating the solution quality. Hence, three pairs of *P_C_* and *P_M_* within their respective value range are selected for comparison: *P_C_* = 0.9, *P_M_* = 0.1; *P_C_* = 0.8, *P_M_* = 0.05; and *P_C_* = 0.7, *P_M_* = 0.01. Since GA is a stochastic search method, the comparative results of 100 Monte-Carlo simulations using the MDIGA for two numbers of planned points are listed in Table 2. The standard deviation is calculated to show how far the set of data are spread out from their average value, which reflects the robustness of algorithm. Under the same working condition, a lower value of standard deviation indicates a better algorithm robustness. In addition, the critical number of iterations (*N_cri_*) at which the solution reaches a convergence level is also presented. In general, minor differences can be found for the three pairs of *P_C_* and *P_M_* in the mean value of optimal planned path, the standard deviation, and the critical number. Only the case of *P_C_* = 0.7, *P_M_* = 0.01 has a relatively larger standard deviation and critical number than the others. Hence, the pair of *P_C_* = 0.9, *P_M_* = 0.1 is sorted for the following study.

Furthermore, three TSPLIB instances are used to compare the feasibility of proposed GAs with some other state-of-the-art approaches, including ant colony optimization (ACO), simulated annealing (SA), and particle swarm optimization (PSO). Similarly, one hundred Monte-Carlo simulations are carried out with the statistical results shown in Table 3. By contrast, the MDIGA has certain advantages for the three problems in terms of the planned path length and the standard deviation. For the problem of eil51, the MDIGA optimizes the path with a length of 448.41 m, which is comparable with the known optimal solution of 426 m.

## 4. Algorithms Evaluation

This section employs Monte-Carlo simulations to evaluate the effectiveness of improved algorithms for the TSP in terms of the number of planned points, the population size, and the computing efficiency. In order to avoid the effects of computer models on the running capacity of algorithms, all the simulations are performed on the same personal computer (Intel (R) Core (TM) i7-7700HQ CPU @ 2.80 GHz) with a memory capacity of 8.0 GB. All algorithms have been coded in MATLAB.

### 4.1. Comparative Results with Various Numbers of Planned Points

Five sample instances from TSPLIB are considered: burma14, ulysses22, eil51, eil76, and rat99. Correspondingly, the five numbers of planned points (*P*) are 14, 22, 51, 76, and 99, with the maximum numbers of iterations (*N_max_*) set as 100, 200, 1600, 2000, and 2000, respectively. In addition, the population size (*S*) is 100. The crossover probability (*P_C_*) and the mutation probability (*P_M_*) in this section are defined as 0.90 and 0.10, respectively. The Monte-Carlo simulations are then repeated one hundred times to obtain the data set of optimal path distances using four algorithms for each TSP instance. The comparative results are presented in box-and-whisker plots (Figure 7), with detailed statistics listed in Table 4.

For each algorithm in every box plot, a range bar is drawn to represent the interquartile range (IQR) of the data set, which indicates the degree of dispersion in a data set. The median value and the average value are identified with a red line and the symbol of a plus sign in the bar. In addition, there are whiskers extending around the bar’s sides. The ends of the whiskers stand for the minimum and maximum values, respectively [26].

When there are 14 planned points, the CGA provides solutions with a longer mean distance and a higher degree of data dispersion than the others in Figure 7a, while the three improved algorithms have similar results of 30.9 m for the average optimal path distance. Meanwhile, the median values of the CGA and the DDIGA are smaller than their average values; this means the two algorithms are easier for producing larger data than the others in one hundred repeated simulations.

As *P* increases from 22 to 99, the CGA always obtains the longest planned path and the largest standard deviation, while the MDIGA has the superior performance in both reducing the path distance and improving the robustness. For the case of *P* = 99, the mean distance and the standard deviation of the MDIGA are 1341.81 m and 31.41 m, which are 49.0% and 79.6% smaller than that of the CGA, respectively. Moreover, the SDIGA performs relatively better than the DDIGA in almost all cases except when *P* = 22, which means not all the improvements in this work are effective for the algorithm. Since both SDIGA and DDIGA have the same number of offsprings as the number of parents, there is no essential difference between the single-domain inversion of SDIGA for enough iterations and the double-domain inversion of DDIGA for enough iterations. It also indicates that only by increasing the number of offsprings could the algorithm performance be optimized substantially. Furthermore, a four-number summary of data sets are listed in Table 4, including the worst, the best, the mean, and the standard deviation values of optimal path distances.

### 4.2. Comparative Results with Different Population Sizes

We chose the TSP instance eil51 with 51 planned points as the working condition in this section. Five population sizes (*S*) of 20, 40, 60, 80, and 100 are considered. Besides, the maximum number of iterations (*N_max_*) for each algorithm is set as 1600. The crossover probability (*P_C_*) and the mutation probability (*P_M_*) are still 0.90 and 0.10, respectively. The Monte-Carlo simulations of one hundred times are then conducted using the four algorithms using every population size. Figure 8 consists of five box-and-whisker plots that show the comparative results. Detailed statistics of optimal path distances are listed in Table 5.

As shown in Figure 8a, all the three improved algorithms, especially the SDIGA and the MDIGA, effectively reduce the optimal path distance and improve the algorithm robustness in comparison with the CGA. In addition, the median value is almost coincided with the mean value in each bar; this means all the algorithms could produce uniformly distributed data under the working condition of eil51.

As shown in Figure 8b–e, evident influence appears that the overall optimal distance is further reduced for each algorithm when *S* increases. Although the robustness of every algorithm changes a little due to the population size, no regular tendency could be found. Furthermore, the algorithm with double-domain inversion fails to outperform the SDIGA in both reducing optimal path distance and improving algorithm robustness, which is not in accordance with our supposition as mentioned in Section 2.2. By contrast, the MDIGA is the most advantageous algorithm for the TSP. In the case of *S* = 60, the mean distance and the standard deviation of MDIGA are 451.63 m and 7.72 m, which are 25.8% and 79.2% smaller than that of CGA, respectively. Detailed statistics, including the worst, the best, the mean, and the standard deviation values of optimal path distance are shown in Table 5.

### 4.3. Comparative Results of Computing Efficiency

The results of five TSPLIB instances with different planned points are employed for comparison of computing efficiency in this section. Two main criteria are selected to evaluate the computing efficiency of every algorithm: time consumption and convergence speed. The former refers to the time cost of completing the maximum number of iterations, and the latter means the critical number of iterations (*N_cri_*) at which the solution reaches to convergence level. Figure 9 shows the convergence history of optimal path distance versus iterations for each algorithm. Meanwhile, all the detailed information of solutions for each algorithm is listed in Table 6.

Overall, the path distance of every algorithm is optimized gradually to be shorter with the increase of iterations, then converges to a stable and horizontal level at a critical number (*N_cri_*), and finally reaches the global optimum. With the increase of planned points’ numbers, both the critical number and the time consumption have a rising tendency for each algorithm. By contrast, the curve of the MDIGA is lower than that of the other algorithms during the entire computing process and has a faster convergence speed and a lower critical number. When *P* = 76 for instance, the MDIGA converges at *N_cri_* = 586, which is 63% faster than the CGA, which spends 46% more time to complete the same iterations. It should be noted that the improved algorithms, especially the SDIGA and the MDIGA, scarify the computation time-cost to guarantee the precision of solution and avoid being trapped in the local optimum.

Furthermore, Figure 10 presents the best trajectories of the five TSPLIB instances (burma14, ulysses22, eil51, eil76, rat99) using the MDIGA. The abscissa and ordinate stand for the values of latitude and longitude of every planned point, respectively. The red number is the sequence of randomly generated points. The start point is enclosed in a red rectangle, and the arrows represent the heading of planned path.

## 5. Application to an Unmanned Surface Vehicle

Nowadays, the unmanned surface vehicle (USV) has been utilized worldwide in both civil and military fields due to the benefits of reducing casualty risk and increasing mission efficiency. As one of the core technologies, the path planning problem is of great significance to realize the autonomous navigation for and control of the USV. In this section, the aforementioned algorithms are applied to plan the route for a self-developed USV. As a preliminary study, the present work neglects the factors of wind, current, and waves in the algorithms.

### 5.1. Unmanned Surface Vehicle Model and Multi-Sensor

The USV model, as shown in Figure 11, is self-designed and constructed by the Sea Wolf group of Qingdao University of Science and Technology. It is 1.8 m in length and 0.9 m in width and has five underwater side bodies. Meanwhile, a 48V 45A battery provides power for electrical motors to drive the propeller.

The navigation, guidance, and control (NGC) system is contained in the hull, with guaranteed waterproof capability. It consists of three module subsystems: the navigation data processing subsystem, path planning subsystem, and autopilot subsystem. In the first subsystem, multi-sensors including electronic compass and GPS (in Figure 12a) are employed for acquiring the direction of the bow and the USV’s location data. An ultrasonic weather sensor, produced by AIRMAR^®^ (Model: WeatherStation^®^ PB200, shown in Figure 12b), is used to collect the real-time, site-specific weather and location information. All the voltage signals from the aforementioned multi-sensors are collected by a navigation data acquisition (DAQ) system. The navigation data is stored in real-time along with ship log and status information.

All the information is then processed and passed to the path planning subsystem, where the GAs are applied to generate an optimal trajectory. According to the planned route, the autopilot employs a closed loop controller to determine the heading and the speed of the USV. In addition, a graphical user interface (GUI) program compiled based on the Spring model view controller (MVC) framework is used to process and record all the data in a personal computer. The general packet radio service (GPRS) wireless network is established as the communication unit between the USV and the personal computer, with an effective distance of 5 km and a transmission speed of 1–100 Mbps. The navigation data acquisition system and the GUI program are shown in Figure 13. It is worth mentioning that there are still several challenges when applying the path planning algorithms to the NGC system of the USV. Since the vehicle have the tendency to deviate from the planned trajectory due to the influence of wind, waves, and currents, the corresponding correction of heading is necessary. Meanwhile, the stability of data transmission for the USV needs to be strengthened, especially when far offshore operation is required. Additionally, the dynamic obstacle detection and avoidance needs to be added to the path planning subsystem, which will have certain demands for the precision of multi-sensors, especially in bad environments.

### 5.2. Application Tests

Corresponding to four working conditions, four numbers of planned points are selected randomly in a practical environment near the Qingdao Olympic Sailing Center at Fushan Bay: 15, 25, 35, and 45. Every condition has the same start point of (N 36°03′22.38″, E 120°22′57.06″) in latitude and longitude. For the location coordinates of other planned points, refer to Table A1, Table A2, Table A3 and Table A4. The four aforementioned GAs are then employed separately to the USV model for comparison of their effectiveness of path planning. The population size (*S*) is set as 100. The maximum numbers of iterations (*N_max_*) are 150, 250, 350, and 450, respectively, which are dependent on the numbers of planned points. In addition, the crossover probability (*P_C_*) and the mutation probability (*P_M_*) are still 0.90 and 0.10.

The convergence history of optimal path distance versus iterations for each algorithm under four working conditions are shown in Figure 14 with detailed information listed in Table 7. Similar conclusions with Figure 9 could be drawn as follows. The MDIGA is superior among the four compared algorithms. Meanwhile, the advantages of the MDIGA in accelerating the convergence and optimizing the path distance become more obvious with the increase of *P*. When *P* = 45 for instance, the curve of the MDIGA converges at *N_cri_* = 186 and obtains the optimal path distance of 77.1 m, which is 33.1% shorter than that of the CGA. In addition, the DDIGA fails to outperform the SDIGA. The trajectory planned by the DDIGA is slightly longer than that of the SDIGA in most cases, as shown in Figure 14a–c. Furthermore, the three improved algorithms require more time for computation than the CGA under the same *N_max_*. However, the MDIGA is not the most time-consuming algorithm, which indicates a desirable capability of balancing the path optimization and the time consumption.

Figure 15, Figure 16, Figure 17 and Figure 18 present the optimal trajectory of each algorithm under each working condition. When there are 15 planned points in Figure 15, the SDIGA and the MDIGA perform better with fewer corners than the CGA around points 3, 12, and 15. Moreover, the trajectories become more complex with more evident differences in path shape and distance as *P* increases. The trajectories generated by the CGA and DDIGA have different levels of path-crossing phenomena in Figure 16a, Figure 17a,c, and Figure 18a; this would be the reason why a longer route distance is generated when compared with the other algorithms under the same condition. However, at the same time, the advantages of the MDIGA reflect more obviously in reducing the path length effectively and simplifying the path shape, especially when more planned points are considered. The reason behind this may be that a larger number of offsprings and the reservation of the fittest individuals could help avoid the local optimum and converge to the optimal solution.

## 6. Conclusions

Based on the CGA, this paper proposes the concept of multi-domain inversion to increase the number of offsprings for the purpose of enhancing the capability of local search and increasing the probability of generating excellent individuals. Monte-Carlo simulations for several TSPLIB examples are carried out to analyze the feasibility and the effectiveness of the improved algorithms. The algorithms are further applied to the path planning problem of a self-developed USV model. Some concluding remarks are summarized as follows:(1)The MDIGA has the superior performance in reducing the optimal path distance, improving the robustness, and accelerating the convergence. The advantages become more obvious with the increase of planned points’ numbers.(2)The increase of population size could help reduce the path distance. However, its effects on algorithm robustness are irregular.(3)The DDIGA fails to outperform the SDIGA in both reducing optimal path distance and improving algorithm robustness. With the same number of offsprings as the number of parents, there is no essential difference between the single-domain inversion of SDIGA for enough iterations and the double-domain inversion of DDIGA for enough iterations.(4)The improved algorithms scarify the computation time-cost to realize the reduction of the optimal path distance and improvement of the algorithm robustness. By contrast, the MDIGA achieves the most desirable balance.(5)The MDIGA is able to reduce the route length effectively and simplify the path shape, especially when more planned points are considered; this is because generating more offsprings and reserving the fittest individuals help guarantee solution precision and avoid being trapped in the local optimum.

However, generating more offsprings and the second fitness evaluation among them would add certain complexities to the algorithm. Hence, more efforts should be made to optimize the computing time-cost. In addition, future study will also focus on comparing the performance of the different ways of generating more offsprings with different encoding methods. Furthermore, comparative studies with different optimization algorithms for the TSP will be also conducted.

## Figures and Tables

**Figure 1 sensors-19-02640-f001:**
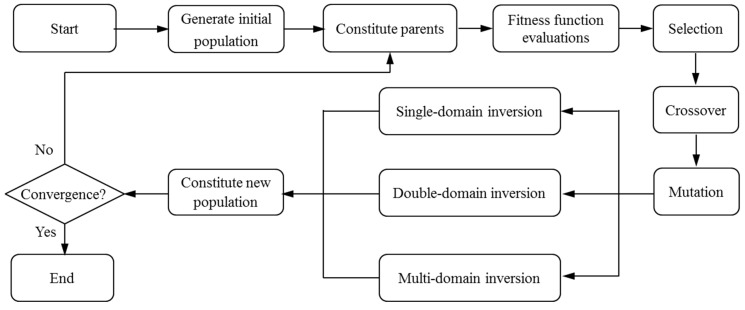
Flow chart of genetic algorithms.

**Figure 2 sensors-19-02640-f002:**
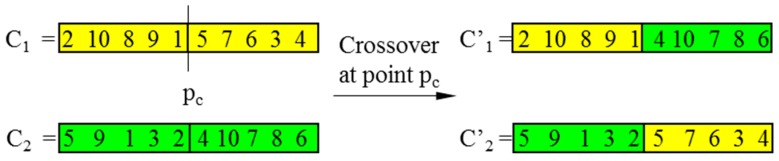
Crossover.

**Figure 3 sensors-19-02640-f003:**
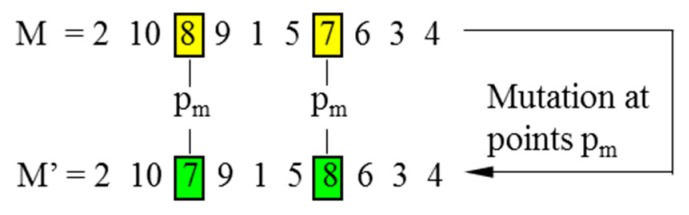
Mutation.

**Figure 4 sensors-19-02640-f004:**
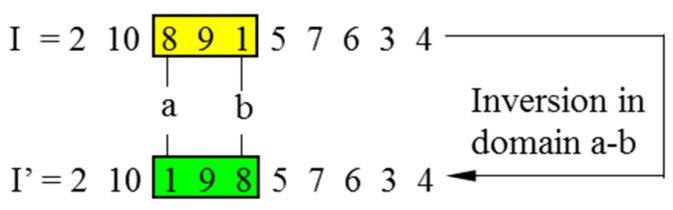
Single-domain inversion.

**Figure 5 sensors-19-02640-f005:**
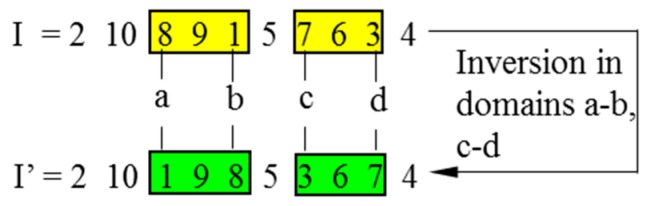
Double-domain inversion.

**Figure 6 sensors-19-02640-f006:**
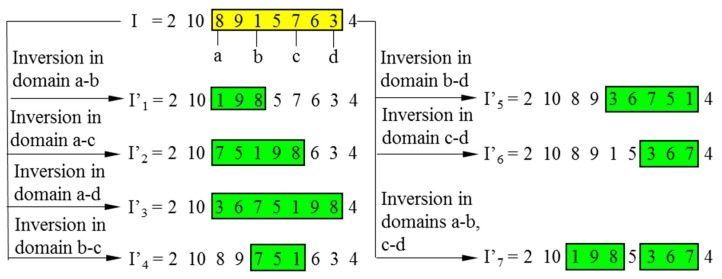
Multi-domain inversion.

**Figure 7 sensors-19-02640-f007:**
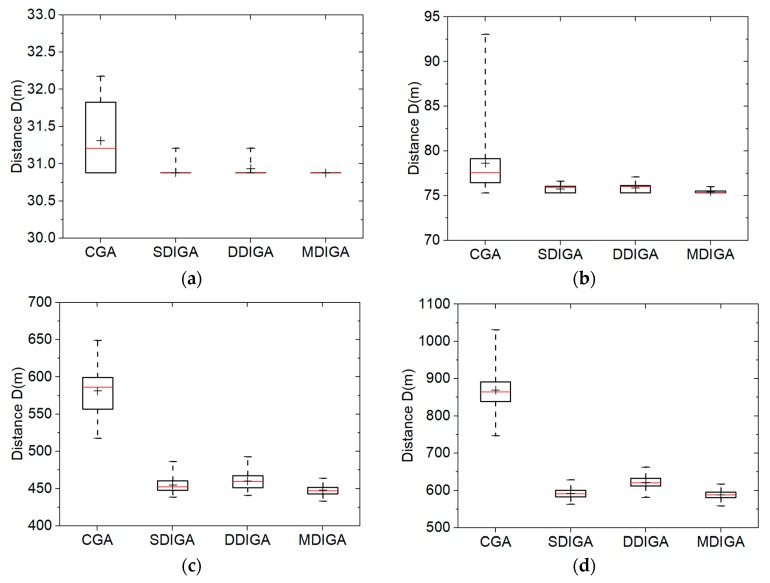
Solution distribution of each algorithm with five numbers of planned points: (**a**) *P* = 14; (**b**) *P* = 22; (**c**) *P* = 51; (**d**) *P* = 76; (**e**) *P* = 99.

**Figure 8 sensors-19-02640-f008:**
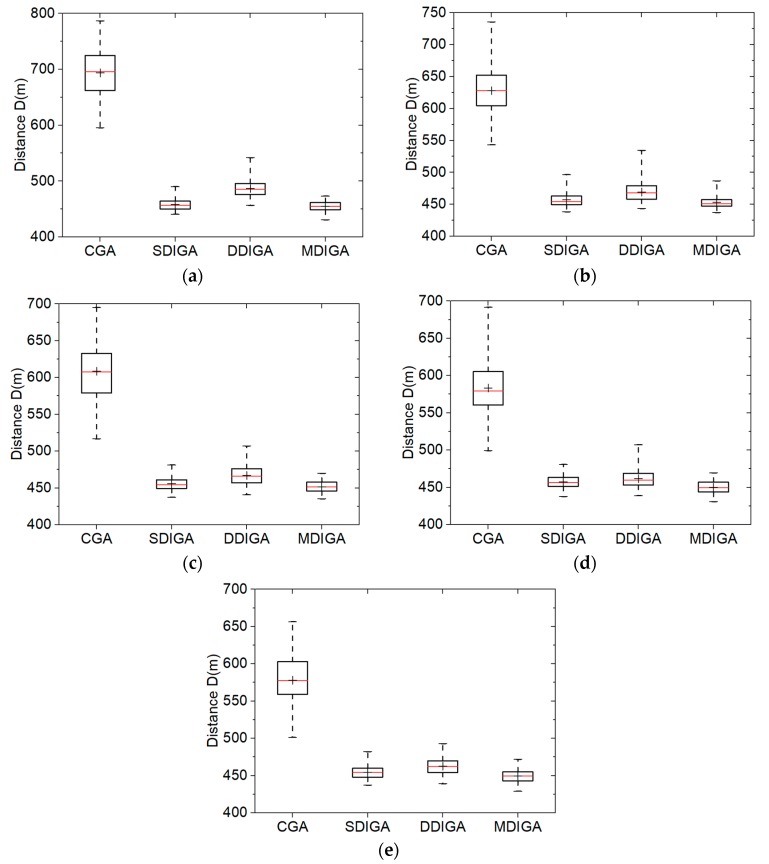
Solution distribution of each algorithm with five population sizes: (**a**) *S* = 20; (**b**) *S* = 40; (**c**) *S* = 60; (**d**) *S* = 80; (**e**) *S* = 100.

**Figure 9 sensors-19-02640-f009:**
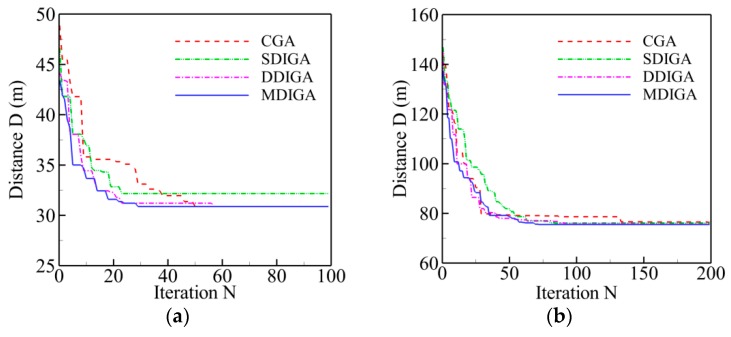
Convergence history of optimal path distance versus iterations for each algorithm: (**a**) *P* = 14; (**b**) *P* = 22; (**c**) *P* = 51; (**d**) *P* = 76; (**e**) *P* = 99.

**Figure 10 sensors-19-02640-f010:**
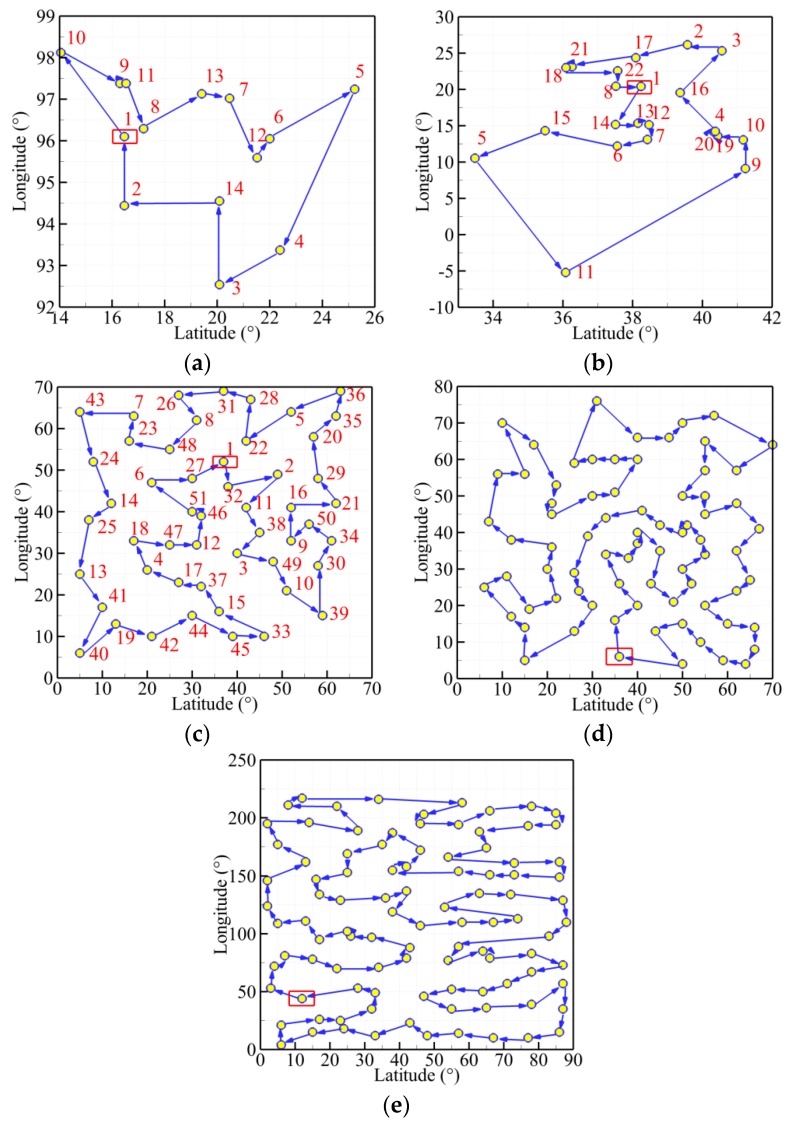
Best routes of the five TSPLIB (library of sample instances for traveling salesman problem) examples: (**a**) burma14; (**b**) ulysses22; (**c**) eil51; (**d**) eil76; (**e**) rat99.

**Figure 11 sensors-19-02640-f011:**
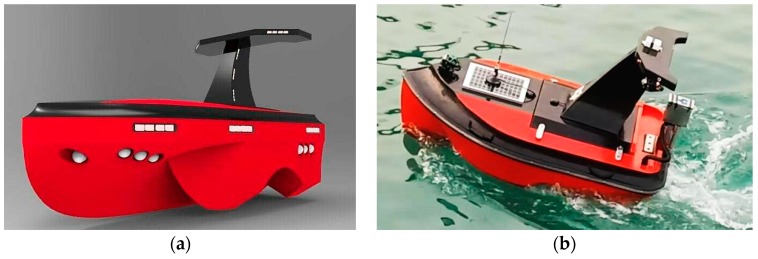
A self-developed unmanned surface vehicle: (**a**) 3D model; (**b**) USV in water.

**Figure 12 sensors-19-02640-f012:**
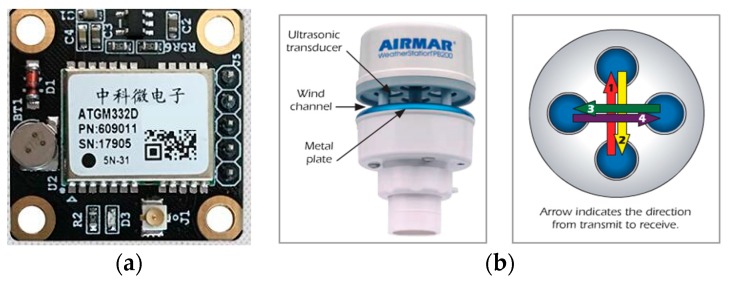
Sensor Equipment: (**a**) GPS; (**b**) ultrasonic transducer.

**Figure 13 sensors-19-02640-f013:**
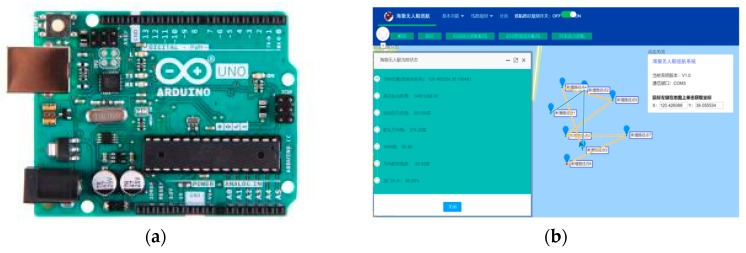
Navigation, guidance, and control system: (**a**) navigation data acquisition system; (**b**) GUI program.

**Figure 14 sensors-19-02640-f014:**
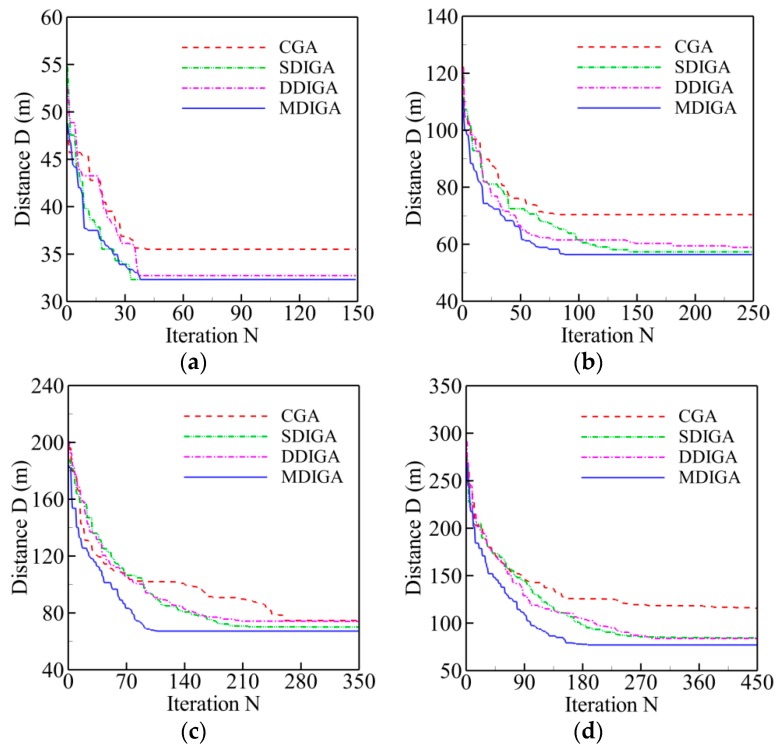
Convergence history of optimal path distance versus iterations for each algorithm: (**a**) *P* = 15; (**b**) *P* = 25; (**c**) *P* = 35; (**d**) *P* = 45.

**Figure 15 sensors-19-02640-f015:**
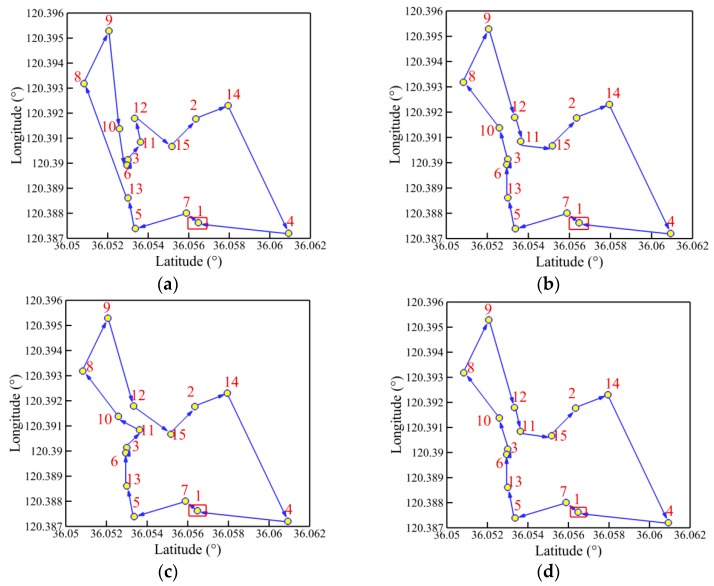
Trajectory planned by each algorithm for *P* = 15: (**a**) CGA; (**b**) SDIGA; (**c**) DDIGA; (**d**) MDIGA.

**Figure 16 sensors-19-02640-f016:**
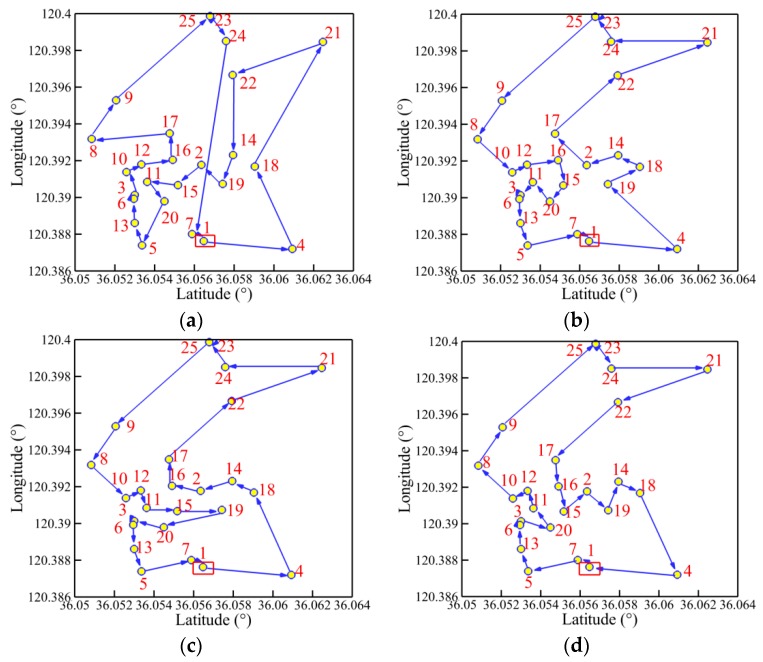
Trajectory planned by each algorithm for *P* = 25: (**a**) CGA; (**b**) SDIGA; (**c**) DDIGA; (**d**) MDIGA.

**Figure 17 sensors-19-02640-f017:**
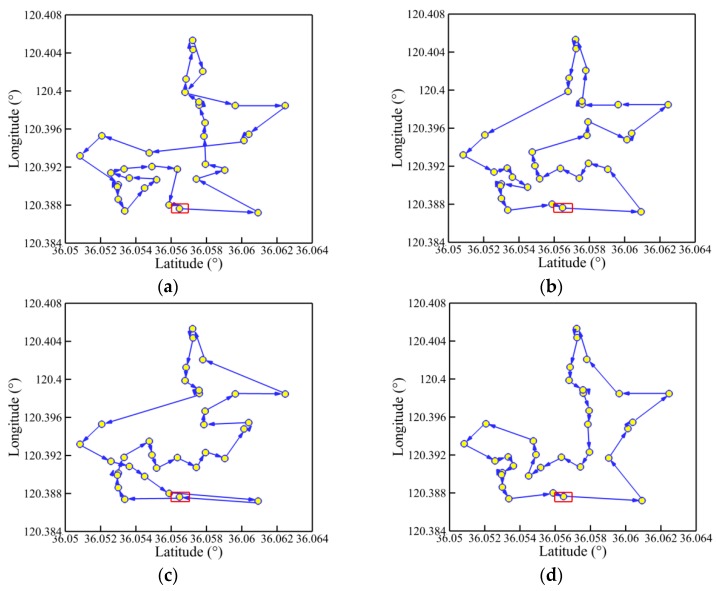
Trajectory planned by each algorithm for *P* = 35: (**a**) CGA; (**b**) SDIGA; (**c**) DDIGA; (**d**) MDIGA.

**Figure 18 sensors-19-02640-f018:**
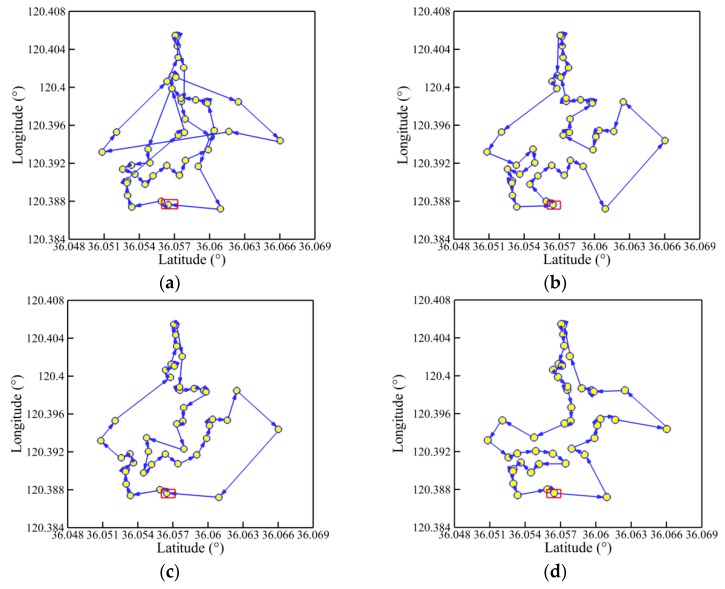
Trajectory planned by each algorithm for *P* = 45: (**a**) CGA; (**b**) SDIGA; (**c**) DDIGA; (**d**) MDIGA.

**Table 1 sensors-19-02640-t001:** Comparative results of planned path length using the multi-offspring method (MO-GA) and multi-domain inversion-based algorithm (MDIGA).

		burma14	eil51	kroB100
Optimal solution	MDIGA	30.88 m	434.08 m	22,918.84 m
	MO-GA	30.88 m	442.19 m	22,141.27 m
Known optimal solution		30.88 m	426 m	22141 m

**Table 2 sensors-19-02640-t002:** Comparative results of various pairs of *P_C_* and *P_M_* for two numbers of planned points.

Case Setting		Best (m)	Mean (m)	Std. Dev. (m)	*N_cri_*
*P* = 51 *S* = 500 *N_max_* = 1600	*P_C_* = 0.9, *P_M_* = 0.10	428.98	443.60	5.84	244
	*P_C_* = 0.8, *P_M_* = 0.05	428.98	441.59	5.10	252
	*P_C_* = 0.7, *P_M_* = 0.01	428.87	443.46	6.11	262
*P* = 76 *S* = 500 *N_max_* = 1600	*P_C_* = 0.9, *P_M_* = 0.10	558.39	575.87	9.07	344
	*P_C_* = 0.8, *P_M_* = 0.05	554.63	573.95	8.84	347
	*P_C_* = 0.7, *P_M_* = 0.01	555.39	576.64	9.98	342

* Std. Dev. is the abbreviation of standard deviation.

**Table 3 sensors-19-02640-t003:** Comparative results with other state-of-the-art approaches for three problems.

Problem	Known Optimal Solution (m)	Algorithm	Worst (m)	Best (m)	Mean (m)	Std. Dev. (m)
ulysses22	75.31	ACO	78.79	75.98	77.74	0.98
		SA	82.57	78.48	81.47	1.17
		PSO	77.10	75.31	75.97	0.46
		DDIGA	77.09	75.31	75.86	0.44
		MDIGA	76.02	75.31	75.46	0.26
eil51	426	ACO	466.92	455.26	459.22	3.41
		SA	478.94	444.96	465.14	8.51
		PSO	480.14	436.06	455.91	9.37
		DDIGA	493.33	441.59	461.03	11.67
		MDIGA	464.82	434.08	448.41	6.38
eil76	538	ACO	599.22	569.61	588.50	8.07
		SA	669.60	634.76	648.68	11.87
		PSO	625.39	564.86	590.11	12.40
		DDIGA	662.56	581.35	621.71	15.03
		MDIGA	617.31	558.39	587.87	11.07

* Std. Dev. is the abbreviation of standard deviation.

**Table 4 sensors-19-02640-t004:** Statistics results of optimal path distance in 100 runs with five numbers of planned points.

*P*	Algorithm	Worst (m)	Best (m)	Mean (m)	Std. Dev. (m)
14	CGA	32.18	30.88	31.31	0.45
	SDIGA	31.21	30.88	30.88	0.03
	DDIGA	31.21	30.88	30.93	0.12
	MDIGA	30.88	30.88	30.88	0.00
22	CGA	93.05	75.31	78.62	3.62
	SDIGA	76.62	75.31	75.76	0.37
	DDIGA	77.09	75.31	75.86	0.44
	MDIGA	76.02	75.31	75.46	0.26
51	CGA	649.91	518.38	581.70	30.13
	SDIGA	486.91	439.13	455.24	9.41
	DDIGA	493.33	441.59	461.03	11.67
	MDIGA	464.82	434.08	448.41	6.38
76	CGA	1031.27	747.25	869.48	47.76
	SDIGA	628.62	563.02	592.26	12.51
	DDIGA	662.56	581.35	621.71	15.03
	MDIGA	617.31	558.39	587.87	11.07
99	CGA	3095.31	2159.52	2635.83	153.87
	SDIGA	1487.05	1293.35	1389.34	38.24
	DDIGA	1670.66	1432.29	1562.88	49.04
	MDIGA	1417.72	1255.06	1341.81	31.41

* Std. Dev. is the abbreviation of standard deviation.

**Table 5 sensors-19-02640-t005:** Statistical results of optimal path distance in 100 runs with five population sizes.

*S*	Algorithm	Worst (m)	Best (m)	Mean (m)	Std. Dev. (m)
20	CGA	786.79	595.42	693.54	43.53
	SDIGA	490.20	440.58	458.17	10.34
	DDIGA	541.94	456.46	486.77	15.31
	MDIGA	473.07	430.75	454.68	8.81
40	CGA	735.63	543.38	628.01	38.08
	SDIGA	496.69	438.34	456.86	11.31
	DDIGA	534.33	443.32	469.31	15.03
	MDIGA	486.53	437.13	453.35	9.34
60	CGA	695.26	516.64	608.39	37.11
	SDIGA	481.26	437.39	455.69	8.95
	DDIGA	507.01	440.81	466.99	13.18
	MDIGA	469.83	435.19	451.63	7.72
80	CGA	691.82	499.26	583.23	33.71
	SDIGA	480.88	437.55	457.38	9.69
	DDIGA	507.26	438.73	461.56	12.22
	MDIGA	469.27	430.75	450.01	8.21
100	CGA	656.51	501.25	577.98	32.48
	SDIGA	482.07	437.03	454.14	8.66
	DDIGA	492.81	439.11	462.64	10.68
	MDIGA	471.82	428.98	449.60	7.84

* Std. Dev. is the abbreviation of standard deviation.

**Table 6 sensors-19-02640-t006:** Simulating results of computing efficiency for each algorithm.

*P*	*N_max_*	Algorithm	*N_cri_*	Time Cost
14	100	CGA	50	1.8
		SDIGA	22	2.6
		DDIGA	57	1.3
		MDIGA	29	2.4
22	200	CGA	133	2.9
		SDIGA	84	4.7
		DDIGA	146	3.1
		MDIGA	72	4.0
51	1600	CGA	697	26.9
		SDIGA	380	42.9
		DDIGA	626	30.4
		MDIGA	377	38.6
76	2000	CGA	1599	39.1
		SDIGA	842	59.4
		DDIGA	1283	43.5
		MDIGA	586	57.2
99	2000	CGA	1870	43.1
		SDIGA	1064	62.9
		DDIGA	1748	47.6
		MDIGA	1002	59.0

**Table 7 sensors-19-02640-t007:** Simulating results of each algorithm with four numbers of planned points.

*P*	*N_max_*	Algorithm	*N_cri_*	Time Cost (s)	D (m)
15	150	CGA	42	1.97	35.50
		SDIGA	33	3.36	32.32
		DDIGA	37	3.12	32.73
		MDIGA	38	5.66	32.32
25	250	CGA	80	3.39	70.37
		SDIGA	142	6.49	57.33
		DDIGA	230	4.01	58.91
		MDIGA	87	5.47	56.38
35	350	CGA	261	4.82	74.67
		SDIGA	283	8.31	70.05
		DDIGA	212	6.47	74.01
		MDIGA	110	7.68	67.20
45	450	CGA	376	6.80	115.19
		SDIGA	382	10.90	84.52
		DDIGA	281	9.94	83.87
		MDIGA	186	10.26	77.07

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
