# Peer review of "An Improved Genetic Algorithm for Path-Planning of Unmanned Surface Vehicle"

_sensors, 2019, doi:10.3390/s19112640_

Reviewer 1 Report

This paper presents interesting methods to improve GA to sol path planning problems. The authors argue that their improvements favor robustness and decrease time for solution convergence. Globally the approach is interesting and shows promising results.

Thereby, I have some concerns with the paper.

(1) In my opinion the similarities and differences of the presented approach with respect to the approach already presented in [17] should be done. Moreover, the results presented should be compared with [17].

(2) Authors argue that their method improve robustness of GA algorithms. Please define what robustness means in your case. Moreover, the results presented have considered only a tuple of Pc and Pm values (Pc=0.9, and Pm=0.1). How could you ensure that your approach will reach the same kind of results for different values for Pc and Pm (given that this values are empirically set). And, how can you also show that your approach will decrease convergence time and increase robustness independently of Pc and Pm values?

(3) The results would also be compared against others state-of-the-art approaches. For example, it is hard to know if the solution reached is better (or not) than others local search algorithms that not exploit GA framework. For instance, heuristic search algorithms (like tabu search), or ant colony algorithms.

(4) Authors claim that MDIGA is able to avoid the intersection of route, simplifying path's shape. But, as far as I understood, there is any mechanism during solution search that ensures it. If is there, any presentation of that have been done. Do it relates to fitting functions used? Could you present them?

(5) the title of the paper refers to "multi-sensor data". But the approach here presented to not relies on multi-sensor data. The quality of localization (precision) do not influences on path planning, given that such a path is computed for a given number of waypoints known a priori. In my point of view, "multi-sensor data" is out of the scope of the paper.

Thus, in my opinion this work should be improved over all these aspects for a further publication.

Minor comments:

- Please add the meaning of TSPLIB abbreviation

- In figures, please replace D by "Distance"

Author Response

We really appreciate all insightful comments and useful suggestions, which definitely will help us to improve the quality and readability of our manuscript. The manuscript has been revised according to the comments. All the textual changes according to the Reviewers’ comments or by the authors ourselves have been shown using the tracking changes. In addition, in order to make the revising contents more clearly, the changes according to the reviewer#1 comments will be highlighted in Blue; the changes according to the editor and reviewer#2 comments will be highlighted in Yellow; the changes according to the common comments and made by the authors ourselves especially for the grammar, spelling and syntax problems will be highlighted in Green

Reviewer 2 Report

The paper proposes an improved Genetic Algorithm and applies to the navigation control of Unmanned Surface Vehicle. In general I find the contributions are marginal, while the comparisons with other GA algorithms are not clearly shown.

The paper shows two improved genetic algorithms by increasing the number of inversion domains and reserving the offspring with the best fitness. This is however unclear to me, since there are numerous GA algorithms and many modified versions to improve the convergence and speeds. In my view the comparisons are not clearly made and the main contributions are vague. 

It will be helpful if the authors can provide some theoretical explanations on why and how the proposed GA algorithms can enhance the performance. If such theoretical analysis is hard to derive, then some intuitive explanations should be added to help the readers.

With the improvement of the proposed GA algorithms, it is not stated what the downsides (trade-offs) are in the algorithms.

The performance is shown by simulations, while the mechanism and theoretical explanations on why Double-Domain Inversion could improve the performance is not shown.

The applications to USV navigation seem to a different topic, while I encourage the authors to emphasize the challenges and new insights on applying the proposed GA algorithms to the navigation (while the current text looks like a direct application).

There are several English errors and writing typos and the authors are encouraged to carefully check the whole paper to improve its presentation. 

Author Response

We really appreciate all insightful comments and useful suggestions, which definitely will help us to improve the quality and readability of our manuscript. The manuscript has been revised according to the comments. All the textual changes according to the Reviewers’ comments or by the authors ourselves have been shown using the tracking changes.In addition, in order to make the revising contents more clearly, the changes according to the reviewer#1 comments will be highlighted in Blue; the changes according to the editor and reviewer#2 comments will be highlighted in Yellow; the changes according to the common comments and made by the authors ourselves especially for the grammar, spelling and syntax problems will be highlighted in Green

Round  2

Reviewer 1 Report

I would like to thank the authors for taking my remarks into account. Paper's quality has been improved, more precisions were added concerning the contribution.
I have read the cover letter and all my questions were addressed.However, there still are some remarks that should be treated.
(1) Concerning my first question about comparison with MO-GA approach. Authors include some comparative results in the cover letter, and the analysis about differences and similarities is very interesting. The MDIGA proposed approach seem to reach good results compared to MO-GA. In my opinion such a comparison may appear on the paper. Even if MDIGA is not always the best, this comparison may be on the paper.
(2) Concerning the values of Pc and Pm probabilities. The citation of [26], the which advises some range values for these probabilities is also welcome. Thereby, the question remains. I completely understand that the linear variation of such parameters could constitute another contribution being revised in another academic journal (this can be also indicate on the paper referring to the submit paper). Anyway, in my opinion, authors would present the constancy on their results for different couples of fixed Pc and Pm parameters (2 couples more at least) in this paper. What is different of the analysis of linear variation of such parameters.
(3) My comment concerning the comparison with the state-of-the-art approaches was more in the sense to indicate the best known solutions for some problems (for instance the symmetric TSPs in http://elib.zib.de/pub/mp-testdata/tsp/tsplib/stsp-sol.html, for which the best solutions are known) in the table. As it, the reader would be able to judge how good the proposed approaches are. In any case, it was interesting to see on the cover letter these comparison, and again I completely understand that a such comparison could constitute a contribution in another academic paper.
(4) The explanation about the fact that MDIGA is able to avoid the intersection, did not convinced me. Could authors formalize on the paper the used fitness functions to evaluate offsprings in order to find the best one? I understand that generating more offsprings and scoring them to keep only the best could help on avoiding intersections, but it does not explains it. I really think that this is intrinsically related with fitness functions.
Minor comments:
- In the introduction, all the references that are cited using the authors name should have also the year. Ex: C. H. Chen et al. -> C. H. Chen et al. (2019)
- C. Silva et al. does not appears on the References section: please check is all cited papers are well referenced
- line 159: a singe -> a single
- line 222: offsprings could the algorithm performance be optimized substancially -> offsprings could substancially optimized algorithm's performance
Author Response

We really appreciate all insightful comments and useful suggestions, which definitely will help us to improve the quality and readability of our manuscript. The manuscript has been revised according to the comments.All the textual changes according to the Reviewers’ comments or by the authors ourselves have been shown using the tracking changes.

Please refer to the wordfile named "Response-Sensors-492022-Round 2-Review1".

Reviewer 2 Report

The paper has been improved by the authors in the revision. However there are certain English typos and writing errors that should  be corrected. 

Some examples:

Page 1: by the implement of---> by implementation, or by using

Page 4: to an USV---> to a USV

Page 8: by human experience???

Page 9, Line 221-222: The sentence has grammatical error.

Line 322: Since the vehicle has--> have

and many others. 

I encourage the authors to carefully revise and check the English of this paper. After that the paper is acceptable. 

Author Response

Response to Reviewers’ Comments- Round 2

We really appreciate all insightful comments and useful suggestions, which definitely will help us to improve the quality and readability of our manuscript. The manuscript has been revised according to the comments.All the textual changes according to the Reviewers’ comments or by the authors ourselves have been shown using the tracking changes.

Responses to the reviewers’ comments are listed as the following:

Reviewer #2

The paper has been improved by the authors in the revision. However there are certain English typos and writing errors that should be corrected.

Some examples:

Page 1: by the implement of---> by implementation, or by using

Page 4: to an USV---> to a USV

Page 8: by human experience???

Page 9, Line 221-222: The sentence has grammatical error.

Line 322: Since the vehicle has--> have

and many others.

I encourage the authors to carefully revise and check the English of this paper. After that the paper is acceptable.

Response: Thanks for the reviewer’s carefully inspection. According to the reviewer’s comments, we have modified the relevant English typos and writing errors.Please check all the track-changings in the revised manuscript named “Sensors-492022-Revised-Round 2”.

Round  3

Reviewer 1 Report

I accept the manuscript in the present form. I also would like to thanks authors, they integrate all m comments and addressed all my remarks. I believe the manuscript proposes a very interesting approach and the quality of the paper is now up coming.